# π2vec: Policy Representation with Successor Features

**Gianluca Scarpellini\* †**
Istituto Italiano di Tecnologia

**Ksenia Konyushkova**
Google DeepMind

**Claudio Fantacci**
Google DeepMind

**Tom Le Paine**
Google DeepMind

**Yutian Chen**
Google DeepMind

**Misha Denil**
Google DeepMind

†: Work done during an internship at Google DeepMind
\*: Corresponding author gianluca.scarpellini@iit.it

## Abstract

This paper introduces **π2vec**, a method for representing black box policies as comparable feature vectors. Our method combines the strengths of foundation models that serve as generic and powerful state representations and successor features that can model the future occurrence of the states for a policy. π2vec represents the behaviors of policies by capturing statistics of how the behavior evolves the features from a pretrained model, using a successor feature framework. We focus on the offline setting where both policies and their representations are trained on a fixed dataset of trajectories. Finally, we employ linear regression on π2vec vector representations to predict the performance of held out policies. The synergy of these techniques results in a method for efficient policy evaluation in resource constrained environments.

## 1 Introduction

Robot time is an important bottleneck in applying reinforcement learning in real life robotics applications. Constraints on robot time have driven progress in sim2real, offline reinforcement learning (offline RL), and data efficient learning. However, these approaches do not address the problem of policy evaluation which is often time intensive as well. Various proxy metrics were introduced to eliminate the need for real robots in the evaluation. For example, in sim2real we measure the performance in simulation (Lee et al., 2021). In offline RL we rely on Off-policy Evaluation (OPE) methods (Gulcehre et al., 2020; Fu et al., 2021). For the purpose of deploying a policy in the real world, recent works focused on Offline Policy Selection (OPS), where the goal is to select the best performing policy relying only on offline data. While these methods are useful for determining coarse relative performance of policies, one still needs time on real robot for more reliable estimates (Levine et al., 2020).

Our proposed π2vec aims at making efficient use of the evaluation time. Efficient offline policy evaluation and selection is relevant in reinforcement learning projects, where researchers often face the challenge of validating improvements. π2vec enables researchers to make more informed decisions regarding which new policy iterations to prioritize for real-world testing or to identify and discard less promising options early in the development process. In particular, we predict the values of unknown policies from a set of policies with known values in an offline setting, where a large dataset of historical trajectories from other policies and human demonstrations is provided. The last step requires policies to be represented as vectors which are comparable and thus can serve as an input to the objective function. Prior work from Konyushova et al. (2021) represents policies by the actions that they take on a set of canonical states, under the assumption that similar actions in similar states imply similar behaviour. However, this assumption is sometimes violated in practice. This work aims at finding more suitable representation by characterizing the policies based on how they change the environment.

To represent policies, our method π2vec combines two components: successor features and foundation models. We adapt the framework of Q-learning of successor features (Barreto et al., 2017) to the

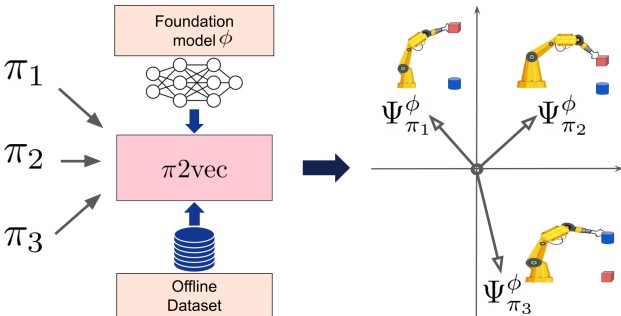

Figure 1: $\pi 2vec$ method relies on the successor feature framework, that we adopt in combination with a dataset of offline demonstrations and a visual foundation model $\phi$. $\pi 2vec$ represents each policy $\pi_i$ as a feature vector $\Psi_{\pi_i}^{\phi} \in \mathbb{R}^n$. $\Psi_{\pi_i}^{\phi}$ encodes the expected behavior of a policy when deployed on an agent.

offline setting by applying the Fitted Q evaluation (FQE) algorithm (Le et al., 2019) which is typically used for off-policy evaluation (OPE). In this work the features for individual states are provided by a general purpose pretrained visual foundation model (Bommasani et al., 2021). The resulting representations can be used as a drop in replacement for the action-based representation used by Konyushova et al. (2021).

Our experiments show that $\pi 2vec$ achieves solid results in different tasks and across different settings. To summarize, our main contributions are the following:

- We propose $\pi 2vec$, a novel policy representation of how the policies change the environment, which combines successor features, foundation models, and offline data;
- We evaluate our proposal through extensive experiments predicting return values of held out policies in 3 simulated and 2 real environments. Our approach outperforms the baseline and achieves solid results even in challenging real robotic settings and out-of-distribution scenarios;
- We investigate various feature encoders, ranging from semantic to geometrical visual foundation models, to show strengths and weaknesses of various representations for the task at hand.

## 2 RELATED WORK

**Representation of black-box policies.** In this paper, our objective is to create vector representations for policies to predict their performance. We treat policies as black-boxes (i.e., no access to internal state, parameters, or architectures) that yield actions for a given observation. It is important to emphasize that our objective differs from representation learning for RL (Schwarzer et al., 2020; Jaderberg et al., 2016; Laskin et al., 2020), as we focus on representing policies rather than training feature encoders for downstream tasks.

Konyushova et al. (2021) studied a setting where the goal is to identify the best policy from a set of policies with a dataset of offline experience and limited access to the environment. Each policy is represented by a vector of actions at a fixed set of states. While this representation performs well in certain applications, it may not be the most effective for predicting policy performance. For instance, consider two policies that generate random actions at each state. These policies do not exhibit meaningfully different beahviour, so for policy evaluation purposes, we expect them to be similar. However, the action policy representation categorizes these policies as different. This paper proposes a method to address this limitation by measuring trajectory-level changes in the environment.

In BCRL (Chang et al., 2022), a state-action feature representation is proposed for estimating policy performance. However, the representation of each policy is independent of other policies and thus cannot be employed to regress the performance of new policies given a set of evaluated policies.

**Offline Policy Evaluation.** Off-policy Evaluation (OPE) aims to evaluate a policy given access to trajectories generated by another policy. It has been extensively studied across many domains (Li et al., 2010; Theocharous et al., 2015; Kalashnikov et al., 2018; Nie et al., 2019). Broad categories of OPE methods include methods that use importance sampling (Precup, 2000), binary classification (Irpan et al., 2019), stationary state distribution (Liu et al., 2018), value functions (Sutton et al., 2016;

Le et al., 2019), and learned transition models (Zhang et al., 2021), as well as methods that combine two or more approaches (Farajtabar et al., 2018). The main focus of the OPEs approaches is on approximating the return values function for a trained policy, while $\pi$2vec goes beyond classical OPE and focuses on encoding the behavior of the policy as vectors, in such a way that those vectors are comparable, to fit a performance predictor.

**Foundation Models for Robotics.** Foundation models are large, self-supervised models (Bommasani et al., 2021) known for their adaptability in various tasks (Sharma et al., 2023). We compare three representative foundation models (Radford et al., 2021; Dosovitskiy et al., 2021; Doersch et al., 2022). Our proposal, $\pi$2vec, is independent of the feature encoder of choice. Better or domain-specific foundation models may improve results but are not the focus of this study.

## 3 METHODOLOGY

### 3.1 OVERVIEW

Our setting is the following. We start with a large dataset of historical trajectories $\mathbb{D}$, and a policy-agnostic state-feature encoder $\phi : \mathbb{S} \to \mathbb{R}^N$. Given a policy $\pi$, our objective is to use these ingredients to create a policy embedding $\Psi_\pi^\phi \in \mathbb{R}^N$ that represents the behavior of $\pi$ (and can be used to predict its performance).

We aim to create this embedding offline, without running the policy $\pi$ in the environment. Although we can evaluate $\pi$ for any state in our historical dataset $\mathbb{D}$, we emphasize that we do not have access to any on policy trajectories from $\pi$, which significantly complicates the process of creating an embedding that captures the behavior of $\pi$.

Our method $\pi$2vec has three steps:

1. Choose a policy-agnostic state-feature encoder $\phi$. We discuss several options for $\phi$ below and in the experiments; however, $\pi$2vec treats the policy-agnostic state-feature encoder as a black box, allowing us to leverage generic state-feature representations in our work.

2. Train a policy-specific state-feature encoder $\psi_\pi^\phi : (\mathbb{S}, \mathcal{A}) \to \mathbb{R}^N$. In this step we combine the policy-agnostic state-feature encoder $\phi$, and the policy $\pi$, to create policy-specific state-feature encoder by training on the historical dataset $\mathbb{D}$. The policy-specific state features $\psi_\pi^\phi(s)$ capture statistics of how $\pi$ would change the environment were it to be run starting from the state $s$.

3. Aggregate the policy-specific state-features to create state-agnostic policy features $\Psi_\pi^\phi$ that represent the behavior of $\pi$ in a state-independent way.

Using the steps outlined above we can collect a dataset of policy-specific state-independent features paired with measured policy performance. This dataset can be used to train a model that predicts the performance of a policy from its features using supervised learning. Because we compute features for a policy using only offline data, when we receive a new policy we can compute its policy-specific state-independent features and apply the performance model to predict its performance **before running it in the environment**. In the following sections we expand on each step.

### 3.2 POLICY-AGNOSTIC STATE FEATURES

The role of the state-feature encoder $\phi$ is to produce an embedding that represents an individual state of the environment. In this paper we focus on state encoders $\phi : I \to \mathbb{R}^N$ that consume single images $I$. Generically our method is agnostic to the input space of the state-feature encoder, but practically speaking it is convenient to work with image encoders because that gives us access to a wide range of pretrained generic image encoders that are available in the literature.

We also consider a few simple ways to construct more complex features from single image features. When each state provides multiple images we embed each image separately and sum the result to create a state embedding. We also consider creating embeddings for transitions $(s, s')$ by computing $\Delta\phi(s, s') \triangleq \phi(s') - \phi(s)$. Both cases allow us to leverage features from pretrained models.

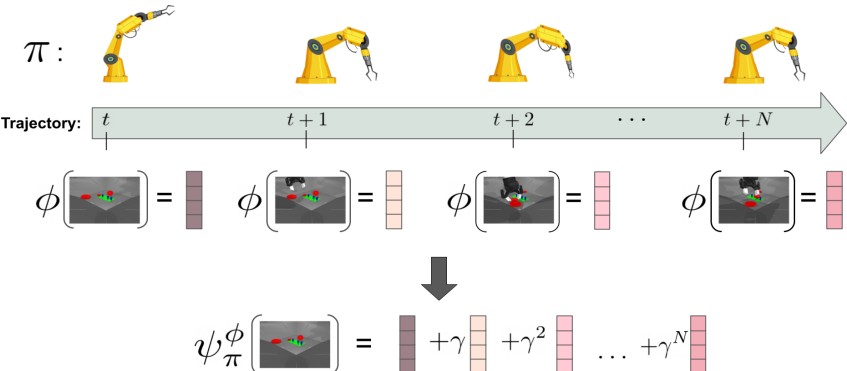

Figure 2: Given a trajectory from the dataset of offline demonstrations, we train successor feature $\psi_\pi^\phi(s_t)$ to predict the discounted sum of features $\sum_i \gamma^i \phi(s_{t+i})$, where $\phi$ is a visual feature extractor and $\pi$ is a policy. Intuitively, $\phi(s_i)$ represents semantic changes in the current state of the environment $s_i$, while successor feature $\psi_\pi^\phi(s_t)$ summarizes all *future* features encoded by $\phi$ if actions came from policy $\pi$.

### 3.3 POLICY-SPECIFIC STATE FEATURES

The next step is to use the policy-agnostic state-feature encoder $\phi$ that provides a generic representation for individual states to train a policy-specific state-feature encoder $\psi_\pi^\phi : (\mathcal{S}, \mathcal{A}) \to \mathbb{R}^N$ that represents the effect that $\pi$ would have on the environment if it were run starting from the given state.

The work of Dayan (1993); Barreto et al. (2017) on successor features provides a basis for our approach to policy representation. We briefly review successor features here, and comment below on how we make use of them. We refer the reader to recent literature covering successor features Lehnert & Littman (2020); Brantley et al. (2021); Reinke & Alameda-Pineda (2021).

Suppose that the reward function for a task can be written as a linear function

$$r(s, a, s') = \langle \phi(s, a, s'), \mathbf{w}_{\text{task}} \rangle, \tag{1}$$

where $\phi(s, a, s') \in R^N$ encodes the state-transition as a feature vector and $\mathbf{w}_{\text{task}} \in R^N$ are weights. Barreto et al. (2017) observe that if the reward can be factored as above, then the state-action-value function for a policy $\pi$ can be written as

$$Q^\pi(s, a) = \mathbb{E}_{(s'|s)\sim D, a\sim\pi(s)} \left[ \sum_{i=t}^{\infty} \gamma^{i-t} r(s_i, a_i, s_{i+1}) \right] = \langle \psi_\pi^\phi(s, a), \mathbf{w}_{\text{task}} \rangle, \tag{2}$$

where

$$\psi_\pi^\phi(s, a) = \mathbb{E}_{(s'|s)\sim D, a\sim\pi(s)} \left[ \sum_{i=t}^{\infty} \gamma^{i-t} \phi(s_i, a_i, s_{i+1}) \right], \tag{3}$$

$(s|s') \sim D$ is a transition from the environment, and $\gamma$ is the discount factor. The corresponding state-value function is $V^\pi(s) \triangleq Q^\pi(s, \pi(s)) = \langle \psi_\pi^\phi(s, \pi(s)), \mathbf{w}_{\text{task}} \rangle \triangleq \langle \psi_\pi^\phi(s), \mathbf{w}_{\text{task}} \rangle$. We will use the notation $\psi_\pi^\phi(s) \triangleq \psi_\pi^\phi(s, \pi(s))$ frequently throughout the remainder of the paper.

The value of $\psi_\pi^\phi(s)$ is known as the *successor features* of the state $s$ under the policy $\pi$. Successor features were originally motivated through the above derivation as a way of factoring the value function of a policy into a task-independent behavior component (the successor features) that is independent of the task, and a task-dependent reward component that is independent of behavior.

For our purposes we will mostly ignore the reward component (although we return to it in one of the experiments) and focus on the behavior term shown in Equation 3. This term is interesting to us for two reasons. First, we can see by inspection of the RHS that the value of $\psi_\pi^\phi(s) = \psi_\pi^\phi(s, \pi(s))$ represents the behavior of $\pi$ as a future discounted sum of state features along a trajectory obtained by running $\pi$ beginning from the state $s$. In other words, $\psi_\pi^\phi$ represents the behavior of $\pi$ in terms of

the features of the states that $\pi$ will encounter, where the state features are themselves given by the policy-agnostic state-feature encoder from the previous section. Figure 2 summarizes the relationship between successor features $\psi$ and state encoders $\phi$.

Second, Equation 3 satisfies the Bellman equation meaning that the function $\psi_\pi^\phi(s, a)$ can be estimated from off-policy data in a task-agnostic way using a modified version of Q-learning, where the scalar value reward in ordinary Q-learning is replaced with the vector valued transition features $\phi(s, a, s')$. We rely on Fitted Q Evaluation (FQE, Le et al. (2019)), an *offline* Q-learning based algorithm, and thus, we obtain a representation of policy behavior purely from data without executing the policy in the environment. Given a dataset $\mathbb{D}$ and a policy $\pi$, FQE estimates its state-action-value function $Q^\pi(s, a)$ according to the following bootstrap loss:

$$L(\theta) = \mathbb{E}_{(s,a,r,s')\sim D, a'\sim\pi(s')} \left[ \|\psi_\theta^\pi(s, a) - (\phi(s, a, s') + \psi_\theta^\pi(s', a'))\|^2 \right]. \tag{4}$$

FQE is simple to implement and it performs competitively with other OPE algorithms in a variety of settings (Fu et al., 2021) including simulated and real robotics domains (Paine et al., 2020; Konyushova et al., 2021). We use FQE with our historical dataset $\mathbb{D}$ to train a policy-specific state-action-feature network $\psi_\pi^\phi(s, a)$, which we then use as the policy-specific state-feature encoder $\psi_\pi^\phi(s) \triangleq \psi_\pi^\phi(s, \pi(s))$ by plugging in the policy action.

### 3.4 STATE-AGNOSTIC POLICY FEATURES

We obtain a single representation $\Psi_\pi^\phi$ of a policy $\pi$ from the state-dependent successor features $\psi_\pi^\phi(s)$ for that policy by averaging the successor features over a set of canonical states:

$$\Psi_\pi^\phi = \mathbb{E}_{s\sim D_{\mathrm{can}}}[\psi_\pi^\phi(s)], \tag{5}$$

where $D_{\mathrm{can}}$ is a set of states sampled from historical trajectories. We sample the canonical states set $D_{\mathrm{can}} \subset \mathbb{D}$ uniformly from from our historical dataset, as in Konyushova et al. (2021), ensuring that each canonical state comes from a different trajectory for better coverage. We average successor features over the same set $D_{\mathrm{can}}$ for every policy. The intuition behind this representation is that $\psi_\pi^\phi(s)$ represents the expected change that $\pi$ induces in the environment by starting in the state $s$; by averaging over $D_{\mathrm{can}}$, $\Psi_\pi^\phi$ represents an aggregated average effect of the behavior of $\pi$.

### 3.5 PERFORMANCE PREDICTION

We aim at predicting the performance of novel, unseen policies. We begin with a dataset of historical policies for which we have measured performance $\Pi = \{\ldots, (\pi_i, R_i), \ldots\}$. For each policy in this dataset we create an embedding using the above procedure to obtain a new dataset $\{\ldots, (\Psi_{\pi_i}^\phi, R_i), \ldots\}$ and then train a performance model $\hat{R}_i = f(\Psi_{\pi_i}^\phi)$ using supervised learning. Given a new policy $\pi_*$ we can then predict its performance before running it in the environment by computing the $\pi$2vec features for the new policy using the above procedure and applying the performance model to obtain $\hat{R}_* = f(\Psi_{\pi_*}^\phi)$.

## 4 EXPERIMENTAL SETUP

In this section we describe the feature encoders, domains, and evaluation procedures, followed by details about our baselines. More details about our architecture, domains, and training procedure can be found in the Appendix.

**Feature encoder.** Firstly, the *Random* feature encoder employs a randomly-initialized ResNet-50 (He et al., 2016). Random features are trivial to implement, and achieve surprisingly strong performance in many settings (Rahimi & Recht, 2007). Here they serve as a simple baseline.

Next, we explore with *CLIP* (Radford et al., 2021). CLIP-network is trained to match image and text embeddings on a large-scale dataset of image caption pairs. Intuitively, by aligning image and text features, CLIP network is trained to encode high-level semantic information.

*Visual Transformers (VIT)* (Dosovitskiy et al., 2021) treat images as a 1D sequence of patches and learn visual features via an attention mechanism. In our experiments the visual transformer is pre-trained on imagenet classification.

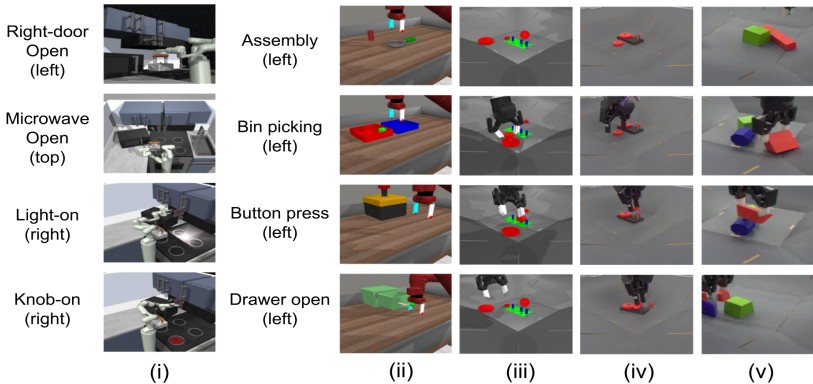

Figure 3: We adopt 5 environments. (i) Kitchen: 5 tasks (Knob-on, Left door open, light on, microwave open, and right door open) and 3 points of views. (ii) Metaworld: 4 tasks (assembly, button press, bin picking, and drawer open) and 3 points of views. (iii) Insert gear in simulation (iii) and (iv) on a real robot. (v) RGB stacking on a real robot.

Lastly, we explore *Track-any-point (TAP)* (Doersch et al., 2022), a general-purpose network for point tracking in videos. The network is pre-trained to track arbitrary points over video sequences and as a result it learns to understand the low-level geometric features in a scene. We use an attention layer trained to select task-relevant features from the TAP model to reduce dimensionality.

This set of feature encoders spans a spectrum of properties as they are created by optimising different objectives. At one extreme CLIP features are trained to align image features with a text description, and encode the semantics of the image. At the other extreme TAP features are trained to track points in videos, and capture low level geometric and texture information. VIT features are in the middle, they need to encode both semantics and local texture to accomplish classification tasks. Depending on the environment and task at hand, better state representation is likely to result in better prediction properties of $\pi$2vec. We leave the question of finding the best representation as future work.

**Domains.** We present extensive experiments to support $\pi$2vec's capabilities across three simulated domains—Insert Gear (Sim), Metaworld, and Franka-Kitchen, and two real domains—Insert Gear (Real) and RGB Stacking (Figure 3). In each domain we use a dataset of offline human demonstrations (Metaworld and Kitchen) and held out policies trajectories (RGBStacking and Insert Gear) for training policy representations. Each policy is treated as a black-box where we do not have any prior knowledge about the architecture or training parameters. We provide further details in Supplementary.

**Evaluation.** We assess the quality of the policy representations by measuring the ability of the model $f$ to predict the performance of held out policies (see Section 3.5). We adopt k-fold cross validation over the set $\Pi$ and report results averaged over cross validation folds. Following previous works on offline policy evaluation (Paine et al., 2020; Fu et al., 2021), we adopt the following three complementary metrics. We report further details in the Supplementary.

- **Normalized Mean Absolute Error (NMAE)** measures the accuracy of the prediction w.r.t. the ground-truth. We adopt MAE instead of MSE to be robust to outliers and we normalize the error to be in range between the return values for each environment. Lower is better.
- **Rank Correlation** measures how the estimated values correlate with the ground-truth. Correlation focuses on how many evaluations on the robot are required to find the best policy. Higher is better.
- **Regret@1** measures the performance difference between the best policy and the predicted best policy, normalized w.r.t. the range of returns values for each environment. Lower is better.

Correlation and Regret@1 are the most relevant metric for evaluating $\pi$2vec on OPS. On the other hand, NMAE refers to the accuracy of the estimated return value and is suited for OPE.

**Baselines.** The problem in this paper is to represent policies in such a way that the representations can be used to predict the performance of other policies given the performance of a subset of policies. Importantly, to address this problem the representation should 1) encode the behavior of the policy, 2) in a way that is comparable with the representations of other policies, and 3) does not require online

Table 1: We compare $\pi$2vec and *Actions* representations for Insert-gear (real) and Insert-gear (sim) tasks, as well as for the RGB stacking environment. The table shows the performance and confidence intervals for different feature representations and encoders.

| Representation | NMAE $\downarrow$ | Correlation $\uparrow$ | Regret@1 $\downarrow$ |
|---|---|---|---|
| | **RGB Stacking** | | |
| *Actions* | $0.261 \pm 0.045$ | $\textbf{0.785} \pm 0.177$ | $0.074 \pm 0.083$ |
| VIT | $\textbf{0.224} \pm 0.063$ | $0.775 \pm 0.146$ | $\textbf{0.036} \pm 0.116$ |
| $\Delta$VIT | $0.344 \pm 0.050$ | $0.030 \pm 0.332$ | $0.375 \pm 0.206$ |
| CLIP | $0.330 \pm 0.042$ | $0.342 \pm 0.293$ | $0.325 \pm 0.180$ |
| $\Delta$CLIP | $0.287 \pm 0.048$ | $0.583 \pm 0.126$ | $0.079 \pm 0.126$ |
| Random | $0.304 \pm 0.066$ | $0.330 \pm 0.334$ | $0.226 \pm 0.177$ |
| $\Delta$Random | $0.325 \pm 0.109$ | $0.352 \pm 0.348$ | $0.190 \pm 0.180$ |
| | **Insert gear (real)** | | |
| *Actions* | $0.252 \pm 0.028$ | $-0.545 \pm 0.185$ | $0.578 \pm 0.148$ |
| Random | $0.275 \pm 0.027$ | $-0.207 \pm 0.267$ | $0.360 \pm 0.162$ |
| CLIP | $\textbf{0.198} \pm 0.030$ | $\textbf{0.618} \pm 0.136$ | $\textbf{0.267} \pm 0.131$ |
| $\Delta$CLIP | $0.253 \pm 0.228$ | $-0.109 \pm 0.100$ | $0.429 \pm 0.100$ |
| | **Insert gear (sim)** | | |
| *Actions* | $0.174 \pm 0.015$ | $0.650 \pm 0.056$ | $0.427 \pm 0.172$ |
| Random | $0.215 \pm 0.026$ | $0.555 \pm 0.104$ | $0.422 \pm 0.143$ |
| TAP | $\textbf{0.164} \pm 0.022$ | $\textbf{0.680} \pm 0.095$ | $0.359 \pm 0.184$ |
| VIT | $0.224 \pm 0.025$ | $0.402 \pm 0.129$ | $0.448 \pm 0.195$ |
| $\Delta$VIT | $0.255 \pm 0.024$ | $0.218 \pm 0.139$ | $0.457 \pm 0.153$ |
| CLIP | $0.180 \pm 0.031$ | $0.502 \pm 0.068$ | $\textbf{0.298} \pm 0.126$ |
| $\Delta$CLIP | $0.189 \pm 0.020$ | $0.586 \pm 0.077$ | $0.314 \pm 0.147$ |

data. Active Offline Policy Selection (AOPS) (Konyushova et al., 2021) stands alone as a notable work that delves into policy representation from offline data with the task of deciding which policies should be evaluated in priority to gain the most information about the system. AOPS showed that representing policies according to its algorithm leads to faster identification of the best policy. In AOPS's representation, which we call "*Actions*", policies are represented through the actions that the policies take on a fixed set of canonical states. We build *Actions* representation as follows. We run each policy $\pi$ on the set of states $D_{\mathrm{can}}$ sampled from historical trajectories. Next, we concatenate the resulting set of actions $\{\pi(s)\}_{s \in D_{\mathrm{can}}}$ into a vector.

To the best of our knowledge, the *Actions* representation is the only applicable baseline in the setting that we adopt in this paper. Nevertheless, OPE methods that estimate policy performance from a fixed offline dataset are standard methodology in offline RL literature. Although these methods do not take the full advantage of the problem setting in this paper (the performance of some of the policies is known) they can still serve for comparison. In this paper, we compared against FQE which is a recommended OPE method that strikes a good balance between performance (it is among the top methods) and complexity (it does not require a world model) (Fu et al., 2021).

## 5 RESULTS

We report results for various feature encoders for Insert gear (sim and real) and RGBStacking. Similarly, we report averaged results for over 4 tasks and 3 point of view for Metaworld and over 5 tasks and 3 point of view for Kitchen. Along with results for each feature encoder, we report the average results of picking the **best** feature encoder for each task (BEST-$\phi$). Similarly, we report as BEST-CLIP and BEST-VIT the average results when adopting the best feature encoder between CLIP/VIT and $\Delta$CLIP/$\Delta$VIT. We identify the **best** feature encoder for a task by conducting cross-validation on previously evaluated policies and pick the best encoder in terms of regret@1.

Our results demonstrate that (i) $\pi$2vec outperforms the *Actions* baseline models consistently across real and simulated robotics environments and multiple tasks, showcasing the framework's effectiveness in representing policies. Furthermore, we demonstrate the applicability to real-world robotic settings, specifically in the challenging Insert Gear (Real) environment, where even underperforming policies contribute to improved policy evaluation. We show that choosing the best model as

Table 2: We evaluate $\pi$2vec on Metaworld and Kitchen. The results are averaged over all settings and confidence intervals are reported. BEST-$\phi$ is $\pi$2vec average performance assuming that we adopt the best $\phi$ in terms of regret@1 for each task-POV setting. Similarly, BEST-CLIP and BEST-VIT are the best feature encoder between CLIP/VIT and $\Delta$CLIP/$\Delta$VIT.

| Representation | NMAE $\downarrow$ | Correlation $\uparrow$ | Regret@1 $\downarrow$ |
|---|---|---|---|
| | **Metaworld** | | |
| *Actions* | $0.424 _{\pm 0.058}$ | $0.347 _{\pm 0.152}$ | $0.232 _{\pm 0.078}$ |
| CLIP | $0.340 _{\pm 0.035}$ | $0.254 _{\pm 0.143}$ | $0.250 _{\pm 0.076}$ |
| $\Delta$CLIP | $0.325 _{\pm 0.092}$ | $0.286 _{\pm 0.154}$ | $0.232 _{\pm 0.086}$ |
| BEST-CLIP | $0.309 _{\pm 0.027}$ | $0.351 _{\pm 0.130}$ | $0.194 _{\pm 0.076}$ |
| VIT | $0.303 _{\pm 0.030}$ | $0.280 _{\pm 0.146}$ | $0.263 _{\pm 0.091}$ |
| $\Delta$VIT | $0.315 _{\pm 0.026}$ | $0.162 _{\pm 0.169}$ | $0.325 _{\pm 0.084}$ |
| BEST-VIT | $0.298 _{\pm 0.029}$ | $0.300 _{\pm 0.147}$ | $0.244 _{\pm 0.092}$ |
| Random | $0.366 _{\pm 0.086}$ | $0.043 _{\pm 0.150}$ | $0.375 _{\pm 0.108}$ |
| BEST-$\phi$ | $\mathbf{0.289} _{\pm 0.018}$ | $\mathbf{0.460} _{\pm 0.099}$ | $\mathbf{0.153} _{\pm 0.060}$ |
| | **Kitchen** | | |
| *Actions* | $0.857 _{\pm 0.128}$ | $0.326 _{\pm 0.128}$ | $0.221 _{\pm 0.089}$ |
| CLIP | $0.417 _{\pm 0.032}$ | $0.021 _{\pm 0.219}$ | $0.317 _{\pm 0.081}$ |
| $\Delta$CLIP | $0.352 _{\pm 0.026}$ | $0.260 _{\pm 0.216}$ | $0.244 _{\pm 0.081}$ |
| BEST-CLIP | $0.333 _{\pm 0.025}$ | $0.346 _{\pm 0.200}$ | $0.197 _{\pm 0.076}$ |
| VIT | $0.385 _{\pm 0.030}$ | $0.030 _{\pm 0.244}$ | $0.322 _{\pm 0.095}$ |
| $\Delta$VIT | $0.344 _{\pm 0.025}$ | $0.155 _{\pm 0.234}$ | $0.251 _{\pm 0.082}$ |
| BEST-VIT | $\mathbf{0.321} _{\pm 0.024}$ | $0.412 _{\pm 0.228}$ | $0.151 _{\pm 0.068}$ |
| Random | $0.382 _{\pm 0.033}$ | $-0.017 _{\pm 0.225}$ | $0.334 _{\pm 0.080}$ |
| BEST-$\phi$ | $0.392 _{\pm 0.053}$ | $\mathbf{0.591} _{\pm 0.203}$ | $\mathbf{0.070} _{\pm 0.045}$ |

a feature-extractor greatly improves results (ii). Finally, we adopt $\pi$2vec to solve Equation 2 and estimate policies' return values in the Metaworld's **assembly** environment, without relying on any ground-truth data (iii). Although the successor feature assumption of linearity of rewards is violated, $\pi$2vec still ranks policies competitively in the offline setting when compared to FQE. In the Appendix, we provide an intuition for choosing the best $\phi$ based on the correlation between task difficulty (iv), and we study the effect of different dataset types, such as demonstrations and trajectories from held out policies (v). We investigate $\pi$2vec's generalization capabilities (vi), including out-of-distribution scenarios (vii). We also demonstrate that $\pi$2vec represents random policies close in the feature space (viii), and that $\pi$2vec is robust to canonical state coverage (ix) and effective with online data (x).

**(i) $\pi$2vec consistently outperforms *Actions*.** We compare $\pi$2vec and *Actions* across all scenarios. Our method outperforms *Actions* representation when predicting values of unseen policies in both real robotics scenarios–RGB stacking and insert-gear (real)–as shown in Table 1. In the former, $\Psi^{\text{VIT}}$ achieves regret@1 of $0.036$ compared to *Actions*' $0.074$, with a relative improvement of $51\%$. In the latter, $\Psi^{\text{CLIP}}$ improves over *Actions* by achieving regret@1 $0.267$ compared to *Actions*' $0.578$ and drastically outperform *Actions* in terms of correlation by achieving $+0.618$ compared to *Actions*' $-0.545$. $\pi$2vec performs robustly on insert gear (real) despite policies' performances for this task vary greatly (see supplementary for per-task policies performances).

We also evaluate our approach in the simulated counterpart *Insert Gear (Sim)*. In this environment, $\Psi^{\text{CLIP}}$ and $\Psi^{\text{TAP}}$ achieve regret@1 of $0.314$ and $0.359$ respectively, compared to *Actions* $0.427$. We underline the dichotomy between geometrical and semantic features: $\Psi^{\text{TAP}}$ performs best in terms of NMAE and Correlation, while $\Psi^{\text{CLIP}}$ outperforms in Regret@1. These results highlight how various $\phi$ compare depending on setting, type of task, and policy performance.

**(ii) When evaluating across multiple settings, selecting $\phi$ leads to better results.** We compare $\pi$2vec with different foundation models across 12 Metaworld settings and 15 Kitchen settings. Table 2 reports the average results across all settings for Metaworld and Kitchen. In Metaworld, we notice that *Actions* performs on par with $\Psi^{\text{CLIP}}$, $\Psi^{\text{VIT}}$, and their respective variations $\Delta$CLIP and $\Delta$VIT, in terms of correlation and regret@1, while our approach consistently outperforms *Actions* in terms of NMAE. As these domains have less state variability, *Actions* represent policies robustly. We test CLIP/$\Delta$CLIP

Table 3: We extend $\pi$2vec to the fully-offline setting and test it on Metaworld assembly task (left, right, and top). We report results and confidence intervals. In this setting, performances of all policies are unknown.

| Representation | NMAE $\downarrow$ | Correlation $\uparrow$ | Regret@1 $\downarrow$ |
|---|---|---|---|
| | **Assembly (left)** | | |
| FQE | $\mathbf{0.338}_{\pm 0.062}$ | $0.125_{\pm 0.218}$ | $0.424_{\pm 0.260}$ |
| $\pi$2vec | $8.306_{\pm 0.155}$ | $\mathbf{0.360}_{\pm 0.097}$ | $\mathbf{0.215}_{\pm 0.079}$ |
| | **Assembly (right)** | | |
| FQE | $\mathbf{0.270}_{\pm 0.093}$ | $-0.029_{\pm 0.351}$ | $0.504_{\pm 0.071}$ |
| $\pi$2vec | $2.116_{\pm 0.056}$ | $\mathbf{0.154}_{\pm 0.115}$ | $\mathbf{0.319}_{\pm 0.080}$ |
| | **Assembly (top)** | | |
| FQE | $\mathbf{0.322}_{\pm 0.012}$ | $-0.251_{\pm 0.516}$ | $0.609_{\pm 0.228}$ |
| $\pi$2vec | $0.492_{\pm 0.006}$ | $\mathbf{0.555}_{\pm 0.106}$ | $\mathbf{0.149}_{\pm 0.071}$ |

and VIT/$\Delta$VIT on previously evaluated policies for each task through cross-validation to identify the best feature encoder for the task in terms of regret@1. We report $\Psi^{\text{BEST-CLIP}}$ and $\Psi^{\text{BEST-VIT}}$ as the average results over the best among CLIP/VIT and $\Delta$CLIP/$\Delta$VIT. $\Psi^{\text{BEST-CLIP}}$ achieves regret@1 0.194 and correlation 0.351, outperforming *Actions* representation. We highlight that the choice of $\phi$ is critical, since $\Psi^{\text{random}}$—using a randomly-initialized ResNet50 as feature extractor—underperforms. Moreover, $\pi$2vec with the best $\phi$ drastically improves, achieving regret@1 of 0.153 compared to *Actions* 0.232. We notice similar improvements when evaluating on Kitchen's 15 settings. Table 2 compares choosing the *BEST* $\phi$ w.r.t. to VIT and CLIP, and against *Actions*. In Kitchen, $\Psi^{\text{VIT}}$ outperforms $\Psi^{\text{CLIP}}$ and *Actions*, while $\Psi^{\text{BEST}-\phi}$ achieves the overall best results.

**(iii) $\pi$2vec enables fully-offline policy selection.** By directly modelling the relationship between successor features and returns, we avoid the linear reward assumption of the original successor features work. This is preferable since rewards are generally not linearly related to state features. However, this restricts our method to settings where some policies' performance is known. To evaluate performance in a fully-offline setting, we fit a linear model the task **reward** $\hat{r} = \langle \phi(s), \mathbf{w}_{\text{task}} \rangle$ given the state's feature representation $\phi(s)$, as in Equation 2 from the original successor features work. Next we predict policies returns as $\hat{R}_i = \langle \Psi^{\phi}_{\pi_i}, \mathbf{w}_{\text{task}} \rangle$. We compare our approach to FQE in Table 3 and find that while our method's return predictions are inaccurate (as evidenced by the high NMAE), it still performs well in ranking policies (higher Correlation and lower Regret@1).

## 6 CONCLUSION

We presented $\pi$2vec, a framework for offline policy representation via successor features. Our method treats the policy as a black box, and creates a representation that captures statistics of how the policy changes the environment rather than its idiosyncrasies. The representations can be trained from offline data, and leverage the pretrained features of visual foundation models to represent individual states of the environment. In our experiments, we represented policies by relying on visual features from semantic (CLIP), geometric (TAP), and visual (VIT) foundation models. We showed that $\pi$2vec outperforms previously used *Actions* based representations and generalizes to fully-offline settings. Overall, our experiments showcase the effectiveness and versatility of $\pi$2vec in representing policies and its potential for various applications in reinforcement learning.

Moving forward, we acknowledge that finding the optimal combination of these elements remains an ongoing challenge. Future work should explore diverse foundation models, offline learning algorithms for successor feature training, and dataset choices. Fine-tuning the feature encoder $\phi$ along with $\psi^{\pi}_{\phi}$ is interesting but pose challenges, as each feature encoder would specialize to predict features for a specific policy, resulting in policy representations that are independent and not comparable. We leave end-to-end fine-tuning as future work. Integrating $\pi$2vec into AOPS framework (Konyushova et al., 2021) for enhanced offline policy selection is another intriguing avenue. Additionally, extending $\pi$2vec to augment the Generalized Policy Improvement (Barreto et al., 2017) in offline settings presents exciting research opportunities.

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

## 7 APPENDIX

### 7.1 DOMAINS

The *Metaworld* Yu et al. (2020) and *Kitchen* Gupta et al. (2019) domains are widely known in the literature. They contain many tasks from multiple views, however, the variability among tasks is low. For example, the robotic arm in Metaworld is initialized within a narrow set of positions, while in Kitchen the object positions are fixed. The task in real robot *RGB Stacking* Lee et al. (2021); Reed et al. (2022) is to stack the red object on top of the blue object with green object as a distractor, where objects are of various geometric shapes. This task is difficult because the objects have unusual shapes and may be positioned at any point in the workspace. We also consider a challenging *gear insertion task* in sim and real where the objective is to pick up a gear of certain size (from an arbitrary position in the workspace) in the presence of other gears and insert it onto a specific shaft on the gear assembly base (arbitrary position in real, fixed in sim). We describe data and policies for each domain below.

**Insert Gear (Sim).**    We use 18 policies for Insert Gear (Sim) task in the simulated environment. We take an intermediate and the last checkpoint for each policy. We collect trajectories with $T = 300$ steps from a single $\pi$ and train all $\psi_{\pi_i}$ on those trajectories. The state $s$ consists of two images, one from a left camera and one from a right camera, and proprioception sensing. All the policies in this domain have the following architecture. Image observations are encoded using a (shared) ResNet, and proprioception is embedded using an MLP. Then, two embeddings are concatenated and further processed by an MLP, followed by an action head.

**Insert Gear (Real).**    The observable state consists of three points of view: a camera on the left of the basket, a camera on the right of the basket, and an egocentric camera on the gripper. The state also includes proprioception information about the arm position. The setup corresponds to the medium gear insertion task described in the work of Du et al. (2023). We conduct experiments on the Insert Gear (Real) task on a real robotic platform by evaluating 18 policies. We collect a dataset of trajectories with a hold-out policy trained on human demonstrations. Next, we train our set of policies on this dataset and we evaluate $\pi$2vec. The state and he policy architecture are the same as in Insert Gear (Sim).

**RGB stacking.**    We use 12 policies trained with behavior cloning on a previously collected dataset of demonstrations for RGB stacking task with a real robotic arm. Each policy is trained with a variety of hyperparameters. The state $s$ consists of images from the basket cameras, one on the left and one on the right, and a first person camera mounted on the end-effector, and proprioception sensing. For training $\pi$2vec, we adopted an offline dataset of trajectories. We collected the trajectories by running a policy trained on human demonstrations. Trajectory length varies between 800 and 1000 steps. We built the evaluation dataset $D_{\text{can}}$ by sampling $5,000$ trajectories and then sampling one state from each of them. Policy architecture follows the description in Lee et al. (2021).

**Metaworld.**    For Metaworld, we consider 4 tasks: assembly, button press, bin picking, and drawer open. We use 3 points of views (left, right, and top), as specified in Sharma et al. (2023); Nair et al. (2022). For each task-camera pair, we adopt 12 policies as proposed by Sharma et al. (2023) for the particular task and point of view. The policies vary by the hyperparameters used during training (learning rate, seed, and feature extractor among NFnet, VIT, and ResNet). Next, we train a successor feature network $\psi_{\pi}^{\cdot}$ for each policy $\pi$ on a cumulative dataset of demonstrations for all tasks and views. At evaluation, we build $D_{\text{can}}$ by uniformly sampling one state from each demonstration.

**Franka-Kitchen.**    For Kitchen, we consider 5 tasks: Knob-on, Left door open, light on, microwave open, and door open with 3 points of views: left, right, and top, as provided by previous works (Sharma et al., 2023; Nair et al., 2022). For each task and point of view, we use human demonstrations provided by Fu et al. (2020). We also adopt policies $\{\pi_i\}$ proposed by Sharma et al. (2023). Each policy solves each task using proprioception and image information from a single point of view, and the policies vary by the hyperparameters used during training (learning rate, seed, and feature extractor among NFnet, ViT, and ResNet). Additional details can be found in Sharma et al. (2023).

Table 4 reports mean and standard deviation of the expected return values for the policies under consideration. We highlight that Metaworld and Insert gear have high standard-deviation (standard

Table 4: We report the mean return value and its standard-deviation for the policies for each environment. For Metaworld and Kitchen, we take the average mean and standard-devation across all tasks and points of views. † Number of policies per task-camera pair.

| Environment | N. policies | Return values |
|---|---|---|
| Insert-gear (sim) | 18 | $0.60_{\pm 0.51}$ |
| Insert-gear (real) | 18 | $2.29_{\pm 1.71}$ |
| RGB Stacking | 12 | $179.41_{\pm 8.21}$ |
| Metaworld | 12† | $215.88_{\pm 164.99}$ |
| Kitchen | 12† | $-47.86_{\pm 20.05}$ |

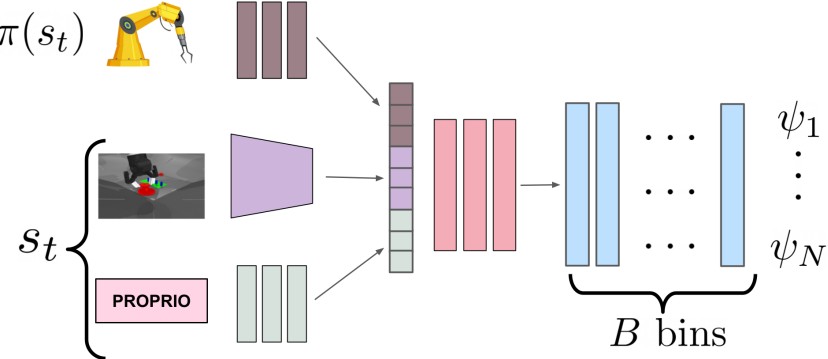

Figure 4: We implement $\psi_\pi^\phi$ as a neural-network. First, we encode state $s_t$–consisting observations and proprioception–and policy actions $\pi(s_t)$ into feature vectors. Next, we concatenate the features and input the resulting vector to a multi-layer perceptron. $\psi_\pi^\phi$ outputs a vector of $B \times N$ dimensions, where $B$ number of bins of the distribution and $N$ is the dimension of the feature vector $\phi(s_t)$. We reshape the output into a matrix, where each row $i$ represents a histogram of probabilities of size $B$ for the successor feature $\psi_i$.

deviation is $75\%$ or more of the mean return value), as we have extremely good and extremely bad policies. On the contrary, return values for RGB Stacking and Kitchen vary less, i.e., most of the policies for these environments achieve similar performance.

## 7.2 ARCHITECTURE AND TRAINING DETAILS

The architecture of successor features network $\psi_\pi^\phi(s)$ for a policy $\pi$ is illustrated in Figure 4. The network takes state-action pairs as input; it encodes actions and proprioception with a multi-layer perceptron, and visual observations using a ResNet50. When the environment observation consists of multiple images (e.g., multiple camera views of the same scene), we embed each image separately and average the resulting vectors. We concatenate the state and action encodings and process the resulting feature vector with a MLP. Finally, the network $\psi_\pi^\phi(s)$ outputs a vector of dimension $R^{N \times B}$, where $N$ is the dimension of the feature vector $\phi(s)$ represented as a distribution over $B$ bins[1]. $\psi_\pi^\phi(s)$ returns $N$ histograms, where each histogram $\psi_i$ approximates the distribution of the discounted sum of feature $\phi_i$ over policy $\pi$. For each environment, we inspect the range of values assumed by $\phi$ to find the min (lower) and max (upper) bound of the histogram. At inference time, we take the expected value of each histogram to compute the successor features vector.

We train $\psi_\pi^\phi(s, a)$ using an FQE with a distributional objective (Le et al., 2019; Bellemare et al., 2017). Training the successor features network only requires offline data (separate for each domain) and does not require any online interactions. We train the network for $50,000$ steps for Metaworld and Kitchen and $100,000$ steps for RGB Stacking, Insert Gear (Sim), and Insert Gear (Real). We adopt

---

[1]We use $B = 51$ bins in all experiments.

Table 5: We conduct additional experiments on Metaworld environment when using a dataset of trajectories for training $\pi$2vec. As expected, enhancing the dataset leads to better performances. We report as BEST-CLIP and BEST-VIT the average results when adopting the best feature encoder between CLIP/VIT and $\Delta$CLIP/$\Delta$VIT in terms of regret@1.

| Representation | Dataset | NMAE | Correlation | Regret@1 |
|---|---|---|---|---|
| *Actions* | - | $0.424 _{\pm 0.058}$ | $0.347 _{\pm 0.152}$ | $0.232 _{\pm 0.078}$ |
| $\Delta$VIT | trajectories | $0.296 _{\pm 0.024}$ | $0.399 _{\pm 0.128}$ | $0.214_{\pm 0.064}$ |
| $\Delta$CLIP | trajectories | $0.278 _{\pm 0.014}$ | $0.469 _{\pm 0.096}$ | $0.189 _{\pm 0.075}$ |
| BEST-$\phi$ | trajectories | $0.269_{\pm 0.017}$ | $0.507 _{\pm 0.105}$ | $0.187 _{\pm 0.073}$ |
| $\Delta$VIT | best | $0.322 _{\pm 0.029}$ | $0.447_{\pm 0.126}$ | $0.191_{\pm 0.074}$ |
| $\Delta$CLIP | best | $0.274 _{\pm 0.026}$ | $0.537 _{\pm 0.166}$ | $0.177 _{\pm 0.175}$ |
| BEST-$\phi$ | best | $\mathbf{0.231} _{\pm 0.016}$ | $\mathbf{0.615}_{\pm 0.086}$ | $\mathbf{0.135} _{\pm 0.052}$ |

different training steps because Metaworld and Kitchen are more predictable than RGB Stacking and Insert gear and less demonstrations are provided. We use the Adam optimizer (Kingma & Ba, 2015) with a learning rate of $3e^{-5}$ and a discount factor of $\gamma = 0.99$. For evaluation, we adopt 3-fold cross-validation in all experiments.

## 7.3 METRICS

We adopt three common metrics Fu et al. (2021): NMAE, correlation, regret@1.

- **Normalized Mean Absolute Error (NMAE)** is defined as the difference between the value and estimated value of a policy:

$$\text{NMAE} = |V^\pi - \hat{V}^\pi|, \tag{6}$$

  where $V^\pi$ is the true value of the policy, and $\hat{V}^\pi$ is the estimated value of the policy.

- **Regret@1** is the difference between the value of the best policy in the entire set, and the value of the best predicted policy. It is defined as:

$$\text{Regret@1} = \max_{i \in 1:N} V_i^\pi - \max_{j \in (1:N)} V_j^\pi. \tag{7}$$

- **Rank Correlation** (also Spearman's $\rho$) measures the correlation between the estimated rankings of the policies' value estimates and their true values. It can be written as:

$$\text{Corr} = \frac{\text{Cov}(V_{1:N}^\pi, \hat{V}_{1:N}^\pi)}{\sigma(V_{1:N}^\pi)\sigma(\hat{V}_{1:N}^\pi)}. \tag{8}$$

Correlation and regret@1 are the most relevant metrics for evaluating $\pi$2vec on Offline Policy Selection (OPS), where the focus is on ranking policies based on values and selecting the best policy.

Regret@1 is commonly adopted in assessing performances for Offline Policy Selection, as it directly measures how far off the best-estimated policy is to the actual best policy.

Correlation is relevant for measuring how the method ranks policies by their expected return value. By relying on methods that achieve higher correlation and thus are consistent in estimating policy values, researchers and practitioners can prioritize more promising policies for online evaluation.

On the other hand, NMAE refers to the accuracy of the estimated return value. NMAE is especially significant when aiming at estimating the true value of a policy and is suited for Offline Policy Evaluation (OPE), where we are interested to know the values of each policy. We assess $\pi$2vec's representation in both settings, showing that $\pi$2vec consistently outperforms the baseline in both metrics. We improve the discussion on metrics in the Appendix of the manuscript.

## 7.4 ADDITIONAL EXPERIMENTS

**(iv) Correlation between task difficulty and $\phi$.** We notice that policy performance varies widely in Insert Gear (Sim) and Insert Gear (Real), as most of the policies fail to solve the task (see

Table 6: We evaluate $\pi$2vec and *Actions* baseline when comparing intermediate checkpoints for a set of policies for Metaworld assembly task and left point-of-view. We highlight that *Actions*representations are similar in such scenario, leading to poor generalization when evaluating on unseen and potentially different policies.

| Representation | NMAE | Correlation | Regret@1 |
|---|---|---|---|
| *Actions* | 0.724 $\pm$0.268 | -0.189 $\pm$0.137 | 0.034 $\pm$0.014 |
| $\Delta$CLIP | **0.570** $\pm$0.173 | **0.190** $\pm$0.361 | **0.029** $\pm$0.034 |

Table 7: We conduct an experiment in an out-of-domain setting. We represent a set of policies that were trained for Metaworld assembly task from *right* point-of-view with *Actions* and $\Psi^{\Delta\text{CLIP}}$. Next, we evaluate how those representations predict the return values if policies are fed with images from a different point-of-view (left camera in our experiment). $\Psi^{\Delta\text{CLIP}}$ outperforms *Actions*in terms of regret@1 and NMAE, supporting our intuition that $\pi$2vec is robust in an out-of-distribution setting.

| Representation | NMAE | Regret@1 |
|---|---|---|
| *Actions* | 0.363 $\pm$0.055 | 0.475 $\pm$0.100 |
| $\Delta$CLIP | **0.227** $\pm$0.078 | **0.300** $\pm$0.106 |

Table 8: We intuitively expect that $\pi$2vec represents random policies close in feature space w.r.t. *Actions* representations, as random policies do not change the environment in any meaningful way. We conduct an experiment with random policies for Metaworld assembly-left to provide quantitative evidence supporting our interpretation.

| Representation | Average distance | Max distance |
|---|---|---|
| *Actions* | 0.39 | 0.62 |
| Random | 0.11 | **0.17** |
| $\Delta$CLIP | **0.03** | 0.22 |

supplementary for per-task policies performances). The gap is evident when compared to the average return value for RGB Stacking, where standard deviation is negligible. Our intuition is that in hard-to-solve scenarios the actions are often imperfect and noisy. This interpretation would explain poor performance of the *Actions* baseline. The comparison of $\Psi^{\text{CLIP}}$ and $\Psi^{\text{VIT}}$ across environments suggests a correlation between the choice of $\phi$ and policies return values. $\Psi^{\text{CLIP}}$ performs better than $\Psi^{\text{VIT}}$ in Insert Gear (Sim), Insert Gear (Real), and Metaworld, where we report the highest standard deviation among policies performances. $\Psi^{\text{CLIP}}$ is robust when the task is hard and most of the policies fail to solve it. On the other hand, $\Psi^{\text{VIT}}$ is the best option in Kitchen and RGB stacking, where the standard deviation of policies returns is low or negligible.

**(v) Studying the performance of $\pi$2vec with different datasets.** We investigate how modifications of the dataset for training $\pi$2vec improves performance in Metaworld. Intuitively, if the training set for $\pi$2vec closely resembles the set of reachable states for a policy $\pi$, solving Equation 2 leads to a more close approximation of the real successor feature of $\pi$ in $(s, a)$. We empirically test this claim as follows. We collect $1,000$ trajectories for each task-view setting using a pre-trained policy. Next, we train successor features network $\psi_\pi^\phi$ for each policy $\pi$ and feature encoder $\phi$, and represent each policy as $\Psi_\pi^\phi$. Table 5 reports results on Metaworld when training $\pi$2vec with the aforementioned dataset (named *trajectory* in the Table). In this setting, $\Psi^{\text{CLIP}}$ and $\Psi^{\text{VIT}}$ outperform both their counterpart trained on demonstrations and *Actions* representation, reaching respectively regret@1 of 0.189 and 0.187. These results slightly improve if we assume to opt for the best dataset for each task and take the average, as reported under *best* dataset in Table 5. Overall, choosing the best feature encoder $\phi$ and best dataset for any given task leads to the best performing $\Psi^{\text{BEST}-\phi}$ with correlation 0.615 and regret@1 0.135–improving over *Actions* by 0.26 and 0.10 respectively.

**(vi) $\pi$2vec generalizes better while *Actions* works with policies are very similar.** We explore how *Actions* and $\pi$2vec compare in the special scenario where all policies are similar. We take 4 intermediate checkpoints at the end of the training for each policy as a set of policies to represent.

Table 9: We evaluate $\pi 2$vec when trained on online data for *Insert Medium (gear)*. Results for $\pi 2$vec from offline data on this task are reported in Table 1.

| Representation | NMAE | Correlation | Regret@1 |
|---|---|---|---|
| *Actions* | 0.174 | 0.650 | 0.427 |
| CLIP | **0.158** | 0.627 | 0.288 |
| Random | 0.348 | 0.425 | 0.302 |
| TAP | 0.172 | **0.686** | **0.205** |

Our intuition is that intermediate checkpoints for a single policy are similar to each other in how they behave. Next, we represent each checkpoint with $\Psi^{\text{CLIP}}$ and *Actions*. We compare cross-validating the results across all checkpoints w.r.t. training on checkpoints for 3 policies and testing on checkpoints for the holdout policy. Table 6 reports results of this comparison on Metaworld's assembly-left task. We notice that *Actions* representations fail to generalize to policies that greatly differ from the policies in the training set. Fitting the linear regressor with *Actions* achieves a negative correlation of $-0.189$ and regret@1 $0.034$. On the other hand, $\Psi^{\text{CLIP}}$ is robust to unseen policies and outperforms Actions with positive correlation $0.190$ and lower regret of $0.029$.

**(vii) $\pi 2$vec performs in out-of-distribution scenarios.** We conduct another investigation to explore $\pi 2$vec performances in a out-of-distribution setting. We hypothesize that $\pi 2$vec represents policies in meaningful ways even when those policies are deployed in settings that differ from the training set, thanks to the generalisation power of foundation models. Table 7 compares $\Delta$CLIP and *Actions* in evaluating policies trained for Metaworld's assembly-right and tested in Metaworld's assembly-left. $\pi 2$vec achieves reget@1 of $0.300$ and NMAE of $0.227$, outperforming *Actions* by $0.175$ and $0.136$ respectively. We leave further exploration of $\pi 2$vec in out-of-distribution settings for further works.

**(viii) $\pi 2$vec represents random policies close in the representation space.** Intuitively, we expect that random policies do not modify the environment in a meaningful way. Therefore, their representations should be closer to each other compared to the similarity between the more meaningful trained policies. We investigate this claim as follows. We provide a set of 6 trained policies and a set of 6 random policies for Metaworld assembly-left. We compute the average and max distance among the random policies representations, normalized by the average intraset distance between trained policies representations. We compare our $\Psi^{\text{CLIP}}$ and $\Psi^{\text{random}}$ with *Actions*. Table 8 reports the results that clearly support our intuition. Both $\Psi^{\Delta\text{CLIP}}$ and $\Psi^{\text{Random}}$ represent random policies close to each other, as the average distance of their representation is respectively $0.03$ and $0.11$ and the maximum distance $0.22$ and $0.17$ respectively. On the other hand, if we represent policies with *Actions*, the representations average and maximum distances are $0.39$ and $0.62$, meaning that random policies are represented far apart from each other.

**(ix) Canonical state coverage.** We sample canonical states uniformly from the dataset that was used for training offline RL policies. Even though there is some intuition that selecting canonical states to represent the environment better can be beneficial, even simple sampling at random worked well in our experiments. We conduct further experiments to ablate the state coverage. We adopt demonstrations from Metaworld Assembly task and adopt the initial state of each trajectory for $\pi 2$vec's and *Actions* representation. By adopting the initial state of a trajectory, $\pi 2$vec cannot rely on the state coverage. We report the results in Table 10. We show that $\pi 2$vec is robust to state coverage, showing SOTA performances even when the canonical states coverage is limited to the first state of each demonstration.

**(x) $\pi 2$vec from online data.** We ideally want to evaluate policies without deployment on a real robot, which is often time-consuming and can lead to faults and damages. Nonetheless, we explore $\pi 2$vec capabilities for representing policies from online data. For each policy $\pi$, we collect a dataset of trajectories by deploying the policy on the agent. Next, we train $\psi_\pi$ on the dataset of $\pi$'s trajectories and compute its $\pi 2$vec's representation. Table 9 reports results when training $\pi 2$vec on online data on Insert Gear (sim) task. We show that $\pi 2$vec's performances improve with respect to the offline counterpart. This result is expected: a better dataset coverage leads to improved results, as we also showed in **(v)**.

Table 10: We test $\pi$2vec robustness to canonical states coverage. We estimate $\pi$2vec's representations for Metaworld's assembly task (left, right, and top point-of-views) by using only the first state of the demonstrations for each task.

| Representation | NMAE ↓ | Correlation ↑ | Regret@1 ↓ |
|---|---|---|---|
| | **Assembly (left)** | | |
| CLIP | 0.381 | 0.359 | 0.287 |
| ΔCLIP | **0.260** | **0.592** | **0.087** |
| Random | 0.422 | 0.252 | 0.46 |
| VIT | 0.366 | 0.26 | 0.347 |
| Actions | 0.356 | 0.503 | 0.222 |
| | **Assembly (right)** | | |
| CLIP | 0.363 | 0.023 | 0.365 |
| ΔCLIP | **0.242** | **0.582** | **0.096** |
| Random | 0.334 | 0.313 | 0.212 |
| VIT | 0.27 | 0.345 | 0.304 |
| Actions | 0.405 | 0.369 | 0.263 |
| | **Assembly (top)** | | |
| CLIP | 0.463 | 0.270 | 0.330 |
| ΔCLIP | **0.305** | **0.594** | **0.078** |
| Random | 0.394 | 0.277 | 0.328 |
| VIT | 0.418 | 0.020 | 0.417 |
| Actions | 0.414 | 0.554 | 0.106 |

