# OpenReview forum: "$\pi$2vec: Policy Representation with Successor Features"
_ICLR.cc/2024/Conference — ICLR 2024 poster_

### Official Review · Reviewer_Kqe9 · 2023-10-31

**Soundness:** 2 fair
**Presentation:** 2 fair
**Contribution:** 2 fair
**Rating:** 5
**Confidence:** 4

**Summary:**

The paper presents an approach to build a compact, vectorial representation of policies, that can be used for off-policy evaluation. The idea is to leverage pre-trained image-based models to generate state-based features to be used for training successor features for policy evaluation. The policy features ($\pi 2vec$) is obtained by averaging the successor features over a set of states sampled from the offline dataset.

The authors showed that these features are effective in the task of off-policy evaluation.

**Strengths:**

The paper is clear and easy to follow. The approach is simple and intuitive, and, as far as I know, it is novel.

**Weaknesses:**

- I would appreciate more details about the relevance of this setting. What are the use cases you have in mind?
- Could you clarify the metrics used to evaluate the approach? You should report (at least in appendix) the equations.
- Why don't you use ranking metrics (Mean Average Precision, DCG, etc)? These seems quite relevant for evaluating methods for offline policy selection. As you mentioned, absolute error in terms of value prediction may not be always relevant.
- You decided to focus on visual representation without clearly explaining why. I think it is important to evaluate the approach also on state-based observation. How would you select base state features in this setting?
- Similarly, I would have expected an evaluation on standard offline benchmarks (e.g., D4RL or ExoRL). For example, if I'm not mistaken, ExORL provides also image-based observations.

**Questions:**

See above.

---

> ### Author Response · Authors · 2023-11-16
>
> Dear reviewer *Kqe9*, thanks for your constructive suggestions. We report our answer about the relevance of this setting and a discussion on the metrics in our general comment, and we answer the remaining questions below.
>
> **Ranking metrics in the experiments:** We add additional details about metrics in our general comment and update the Appendix accordingly. To answer the reviewer’s specific question, Correlation is a ranking metric that evaluates the ranking of the estimated policies’ values, which we clarify in the manuscript. It is commonly adopted for Offline Policy Selection [1] and can be used for comparing policies. We report correlation for all our experiments to assess $\pi$2vec’s performances on ranking policies.
>
> **Visual representations**: Commonly available foundation models are usually trained on visual (e.g., VIT, TAP) and text-vision data (e.g., CLIP). Those features are commonly adopted in robotics tasks [2]. Motivated by the fact that learning RL policies from vision is hard for such a task at hand, we explore how $\pi$2vec for vision-based RL benefits from visual foundation models thanks to their geometrical (TAP), semantical (CLIP), and classification-related (VIT) features. Extending $\pi$2vec for state-based representations requires foundation models that include proprioception or other state-based information as part of their input. Although state-based features are relevant for future work, in this paper, we focused on showing that visual representations are sufficient for policy representations in the domains that we considered.
>
> **Evaluation on standard offline benchmarks**: We are interested in vision-based robotic environments, and we leave further evaluation of $\pi$2vec on different scenarios as future work. [3] is limited in scope to synthetic environments with a focus on state-based RL (e.g., CartPole, Walker), while [4] evaluates a broader range of environments, including synthetic (Maze2D), navigation (CARLA, AntMaze), and robotic manipulation (Kitchen) environments. In our experiments, we included the Kitchen environment along with other real and synthetic environments for vision-based robotics (Metaworld, RGBStacking, Insert gear sim and real)
>
>
> [1] Justin Fu, Mohammad Norouzi, Ofir Nachum, George Tucker, Ziyu Wang, Alexander Novikov, Mengjiao Yang, Michael R Zhang, Yutian Chen, Aviral Kumar, et al. Benchmarks for deep off-policy evaluation. arXiv preprint arXiv:2103.16596, 2021
>
> [2] Mohit Sharma, Claudio Fantacci, Yuxiang Zhou, Skanda Koppula, Nicolas Heess, Jon Scholz, and Yusuf Aytar. Lossless adaptation of pretrained vision models for robotic manipulation. ICLR 2023.
>
> [3] Yarats, Denis, et al. "Don't change the algorithm, change the data: Exploratory data for offline reinforcement learning." arXiv preprint arXiv:2201.13425 (2022).
>
> [4] Fu, Justin, et al. "D4rl: Datasets for deep data-driven reinforcement learning." arXiv preprint arXiv:2004.07219 (2020).

---

### Official Review · Reviewer_eE8e · 2023-10-31

**Soundness:** 3 good
**Presentation:** 3 good
**Contribution:** 3 good
**Rating:** 6
**Confidence:** 4

**Summary:**

This paper proposes a method for representing reinforcement learning policies as comparable feature vectors. The idea is to use successor features to capture how a policy changes the states of the environment over time. In particular, the method leverages pretrained foundation models to encode individual states into feature vectors. After training the successor features on offline datasets of trajectories, the method aggregates the state-dependent successor features into a state-independent policy embedding that summarizes the policy's behavior. Finally, the embeddings are used to predict policy performance.

**Strengths:**

1. Novel policy representation method: The combination of successor features and foundation models provides a new way to summarize and compare policies based on their effects on the environment. Representing policies by their induced state changes rather than their parameters or actions is an interesting idea.

2. Strong empirical results: The paper presents extensive experiments across 5 domains, including real robots. The method outperforms the baseline policy representation method in predicting held-out policy performance.

**Weaknesses:**

1. Limited theoretical analysis: The paper shows empirically that the obtain successor representations are effective, but provides limited insight/intuition or analysis into why combining successor features and foundation models results in good policy embeddings.

2. Restricted to offline setting: The method seems to require pre-collected offline datasets and cannot be applied in an online setting where policies interact with the environment. The offline assumption limits the applicability.

**Questions:**

1. Why do successor features plus foundation models work well for policy representation? More analysis on why this combination is effective compared to other representations would be useful.

2. Can this method be extended to an online setting where policies interact with the environment?

3. How does dataset composition impact the quality of the policy embeddings? Is there any further analysis on dataset requirements and relationships between dataset and representation quality?

4. Have you considered any other methods to aggregate the state-dependent successor features? Averaging seems effective but overly simplistic; are there alternatives that may capture policies better?

---

> ### Author Response · Authors · 2023-11-16
>
> Dear reviewer *eE8e*, thank you for the comments and suggestions to improve our paper. We address the question about online representation in the general comment. We proceed to answer the remaining concerns below.
>
> **Limit analysis on why combining successor features and foundation models is effective:**
> $\pi$2vec is theoretically justified by the successor feature framework [1]. We based the method on the assumption that visual foundation models consistently represent pertinent features of the environment and are adaptable as reward predictors[2]. Our focus is, therefore, on the empirical evaluation of $\pi$2vec’s representations.
> We validate this assumption in paragraph (iii) of the experimental Section when we validate $\pi$2vec for fully offline policy evaluation. In this setting, we fit the reward predictor $\hat{r}=\langle\phi(s), w_\text{task}\rangle$ from offline data, where $\phi$ is a foundation model, and we estimate $V^\pi(s)$ by injecting $w_\text{task}$ in Equation 1 of the main manuscript. Even if this assumption might not be true in general, we show in Table 3 that it is reasonable for the settings we consider. Table 3 reports that estimating value functions with $\pi$2vec representations is a strong offline policy evaluation method compared to FQE, a state-of-the-art method for OPE.
>
> **Comparing feature representations for $\pi$2vec:**
> We clarify our insights about $\pi$2vec representations. We show that $\pi$2vec with foundation models always outperforms random features (see Tab. 1 and 2). We compare geometrical (TAP) and semantical (CLIP) features in Table 1 for the task Insert Gear (sim). We show that geometrical representations improve for correlation, while semantical task helps for Regret@1. Furthermore, we discuss our insight by comparing CLIP and VIT for policy representations in paragraph (iv) of the Appendix. From our analysis, we suggest that $\pi$2vec with CLIP features performs better when the historical policies do not solve tasks effectively. This is often the case when starting working on a new robotic task, e.g., insert gear (real). On the other hand, VIT features help find the best policy among a set of well-performing policies, which is ideal in ablating policies that achieve good performances for the task.
>
> **Dataset quality**: we explore dataset quality in paragraph (v) in Section 7.4 of the Appendix. We collect an enhanced dataset of trajectories for all Metaworld tasks and point-of-views (12 settings). A dataset of trajectories better resembles the behavior of historical policies that are used for $\pi$2vec’s representations. We report results in Table 5, showing that $\pi$2vec improves performance when adopting an improved dataset. Similar findings hold when adopting $\pi$2vec on online data.
> Other aggregation methods: we thank the reviewer for the suggestion. In this initial work, we focus on proposing $\pi$2vec’s representations, and we leave aggregation strategies as further work.
>
> [1] Barreto, André, et al. "Successor features for transfer in reinforcement learning." Advances in neural information processing systems 30 (2017).
>
> [2] Du, Yuqing, et al. "Vision-language models as success detectors." CoLA (2023).

---

### Official Review · Reviewer_4YPE · 2023-11-03

**Soundness:** 2 fair
**Presentation:** 2 fair
**Contribution:** 2 fair
**Rating:** 5
**Confidence:** 3

**Summary:**

This paper proposed a new off-policy evaluation apporach.
The problem setting is that given a set of historical trajectories sampled from other policies and its corresonding performance, it predicts the performance of a unseen policy.

The main idea is using the average of successor features over a set of canonical states of each policy, to predict the policy performance.
The successor features are obtained via Fitted Q learning via a policy-agnositic state encoder.

The authors show that proposed approach is usually better than baseline across different domains and metrics.

**Strengths:**

- This paper studied an important problem of off-policy evaluation.
- As far as I k now, the idea of leveraging successor feature for the purpose of OPE is new.
- The authors conducted extensive set of experiments.

**Weaknesses:**

- Potentially missing baseline: I am not closely follow the OPE literature, but could the authors explain why [1] is not suitable for this setting?
We should be able to get ranking results by that approach.
- Missing key ablations: I think there are implicit key assumptions such as
	- 1) the historical policies' performance should mostly likely cover the unseen policy. If not, the unseen policy's performance is way better, I hardly expect this approach would work.
	- 2) the canonical states coverage. How does it affect the performance if state coverage of MDP is small? More specifically, if the historical dataset only cover 50% of
	states that the unseen policy would visit in the MDP, how does it affect the performance?

	It would be good to have a comprehensive understanding when this approach can be effective.
- In section 3.4, \Phi_\pi represents an aggregated average effect of the behavior of π, I think this highly relies on the implicit assumptions mentioned above. It would be
good that authors can clarify.


[1] Alexander Irpan, Kanishka Rao, Konstantinos Bousmalis, Chris Harris, Julian Ibarz, and Sergey Levine. Off-policy evaluation via off-policy classification. Advances in Neural Information Processing Systems, 32, 2019.

**Questions:**

Please refer to the weakness section.

---

> ### Author Response · Authors · 2023-11-16
>
> Dear reviewer *4YPE*, thank you for the comments and suggestions to improve our paper. We report our answer about when $\pi$2vec is effective by motivating the use cases of $\pi$2vec in the general comment, and we address the remaining concerns below.
>
> **Comment on [1]**: Irpan et al. [1] propose an OPE approach for robotics. Authors assume that reward is sparse and binary (success/failure in an episode), and they cast OPE as a binary classification problem. Moreover, their assumption about sparse and binary rewards does not hold in our setting, as the environments that we adopted have continuous and dense rewards. In experiment (iii) of the main paper, we compare our method to FQE, which is a strong baseline in  OPE (similar to [1] but without the assumption of binary rewards), showing promising results.
> Coverage of historical policies: While it is true that very differently performing policies are unlikely to provide useful information for policy evaluation of the new policies, we have some experiments that look at the question of how the difference in the policies affects the performance of the method. We explore the point raised by the reviewer in paragraph (vi) in the Appendix. We adopt $\pi$2vec for representing intermediate checkpoints for a set of policies for Metaworld assembly task, and we test on held-out policies. Intermediate checkpoints for the same policy are similar to each other and different from checkpoints of other policies, especially at the beginning of training [2].  $\pi$2vec greatly outperforms actions in this setting for all metrics, proving its robustness to policy coverage.
>
> **Canonical state coverage**: We sample canonical states uniformly from the dataset that was used for training offline RL policies. Even though selecting canonical states to represent the environment better is intuitively beneficial, even simple sampling at random worked well in our experiments. We conduct further experiments to ablate the state coverage. We adopt demonstrations from Metaworld Assembly task and adopt the initial state of each trajectory for $\pi$2vec’s and Action’s representation. By using the initial state of a trajectory, $\pi$2vec cannot rely on the state coverage. We report the results in the following table. We show that $\pi$2vec is robust to state coverage, showing SOTA performances even when only initial states are used. We add this experiment in the Appendix of the revised manuscript.
>
> *Assembly Left*
> | Feature  	|	NMAE | Correlation | Regret@1 |
> |--------------|---------|-------------|----------   |
> | CLIP     	|   0.381 |   	0.359 |	0.287 |
> | $\Delta$CLIP | **0.260** | 	**0.592** |  **0.087** |
> | Random   	|   0.422 |   	0.252 | 	0.46 |
> | VIT      	|   0.366 |    	0.26 |	0.347 |
> | Actions  	|   0.356 |   	0.503 |	0.222 |
>
> *Assembly Right*
> | Feature   |	NMAE | Correlation | Regret@1 |
> |-----------|---------|-------------|----------   |
> | CLIP  	|   0.363 |   	0.023 |	0.365 |
> | $\Delta$CLIP      	| **0.242** | 	**0.582** |  **0.096** |
> | Random	|   0.334 |   	0.313 |	0.212 |
> | VIT   	|	0.27 |   	0.345 |	0.304 |
> | Actions   |   0.405 |   	0.369 |	0.263 |
>
> *Assembly Top*
> | Feature  	|	NMAE | Correlation | Regret@1 |
> |--------------|---------|-------------|----------   |
> | CLIP     	|   0.463 |   	0.270 |	0.330 |
> | $\Delta$CLIP | **0.305** | 	**0.594** |  **0.078** |
> | Random   	|   0.394 |   	0.277 |	0.328 |
> | VIT      	|   0.418 |   	0.020 |	0.417 |
> | Actions  	|   0.414 |   	0.554 |	0.106 |
>
> **Interpretation for $\Phi_\pi$**: As mentioned by reviewer *4YPE*, $\Phi_\pi$ represents policy $\pi$ by aggregating the successor feature representation for policy $\pi$ over a canonical set of states. $\pi$2vec performances are influenced by (i) historical policy coverage, (ii) coverage of the training data, and (iii) canonical states. Our experiments show that (i) $\pi$2vec is robust to historical policy coverage (paragraph *vi* and *x*, Section 7.4 in the revised Appendix), (ii) including online trajectories in the training set improves $\pi$2vec’s representations (paragraph *v*, section 7.4 in Appendix), and (iii) canonical state coverage has a limited impact on $\pi$2vec (paragraph ix in the revised manuscript in the Appendix). We hope our answers address the question and are open for further discussion.
>
> [1] Alexander Irpan, Kanishka Rao, Konstantinos Bousmalis, Chris Harris, Julian Ibarz, and Sergey Levine. Off-policy evaluation via off-policy classification. Advances in Neural Information Processing Systems, 32, 2019.
>
> [2] Konyushova, Ksenia, et al. "Active offline policy selection." Advances in Neural Information Processing Systems 34 (2021): 24631-24644.

---

> ### Comment · Reviewer_4YPE · 2023-11-22
> **Further questions**
>
> Thanks for the detailed response from the authors.
>
> > canonical state coverage has a limited impact
>
> I didn't really understand the intuition behind this. To do a thought experiment, if we only hav one state, how could this approach work? I am a bit skeptical on this conclusion.
>
> > even simple sampling at random worked well in our experiments.
>
> I am not sure if using different selection approach to approve the state coverage is a valid reasoning process. it would be great if we know how many unique states are covered, and what is percentage of those `canonical states/full state`, more importantly  `canonical states/full state sampled from high quality policy` I might be missing sth but I couldn't find such info in the paper and author response.

---

> > ### Author Response · Authors · 2023-11-22
> >
> > Dear reviewer *4YPE*, thanks for the questions. We hope our response will further clarify the role of canonical states in our approach.
> >
> > ## Clarification on coverage for the training dataset and canonical state set
> > We train $\psi_\pi^\phi$ via FQE [1] to predict the discounted sum of features $\phi$ for policy $\pi$ on an offline dataset $D$. In this respect, $\psi_\pi^\phi$  has common ground with classical FQE and offline RL in general: if a region of the state space is not represented in the training set $D$ and policy $\pi$ accesses that region at deployment, then $\psi_\pi^\phi$ does not capture any behavior of the unrepresented region. However, this is a limitation of the setting rather than the method and is shared by all offline RL methods [3], as they rely on a static dataset. Nonetheless, we can still attempt to make the best estimate possible given the data available. As we argued in the “use case” paragraph of our general comment, this setting has merit since it allows us to prioritize using scarce evaluation resources (i.e., real robots). Moreover, we would expect $pi$2vec estimates to become better over time, as more data is recorded and thus better coverage of the state space is achieved.
> >
> > ## Improving canonical state sampling
> > For RGBStacking, Insert Gear (sim), and Insert Gear (real), we sample $D_\text{can}$ from the training set of high-quality trajectories $D$ by first randomly sampling 5000 trajectories from $D$ and uniformly sampling one unique state per trajectory. For Metaworld and Kitchen, we build $D_\text{can}$ by sampling one unique state from each human demonstration in $D$. Across all tasks and environments, each state in $D_\text{can}$ is unique and comes from a high-quality (successful) demonstration for the task. Details can be found in Appendix 7.1.
> >
> > We tested various sampling strategies but relied on naive random sampling, as in [2], as we didn’t notice significant improvements. Selecting unique states for $D_\text{can}$ that also guarantees better coverage over the high-reward regions is outside the scope of our work and is an interesting research direction.
> >
> > There are two sources of error in generating features with $pi$2vec (i) the bias in training $\psi_\pi^\phi$ via FQE and (ii) the variance in estimating $\Psi^\phi_\pi = E_{s \sim D_\text{can}}[\psi^\phi_{\pi}(s)]$ on the canonical set $D_\text{can}$. We suspect that the FQE bias is likely the dominant source of error.
> >
> > [1] Le, Hoang, Cameron Voloshin, and Yisong Yue. "Batch policy learning under constraints." International Conference on Machine Learning. PMLR, 2019.
> >
> > [2] Konyushova, Ksenia, et al. "Active offline policy selection." Advances in Neural Information Processing Systems 34 (2021): 24631-24644.
> >
> > [3] Levine, Sergey, et al. "Offline reinforcement learning: Tutorial, review, and perspectives on open problems." arXiv preprint arXiv:2005.01643 (2020).

---

### Official Review · Reviewer_81Fm · 2023-11-08

**Soundness:** 3 good
**Presentation:** 3 good
**Contribution:** 2 fair
**Rating:** 5
**Confidence:** 3

**Summary:**

The ability to represent reinforcement learning policies in a vector space would allow for quickly evaluating policies' performance offline. Successor features is a learning paradigm that defines a policy's performance (i.e., expected reward) as being linear in the features of the policy. The authors propose to leverage ideas from successor features by predicting the performance of unknown policies with the performance of known policies based on joint learned successor feature embedding. This would allow for evaluating unknown policy's performance offline without needing online interactions.

**Strengths:**

- The idea of storing an embedding space where you can look up policies could be beneficial for OPE and imitation learning.
- The combination of exploring foundation model features and successor features is interesting.
- The authors provide a very thorough empirical investigation of the performance of the proposed idea across several tasks. Furthermore, they investigated various representation learning ideas beyond changing the underlying foundational model, which provided insight into which representation is important for each task.
- The evaluation metrics chosen provide insight between relative performance (i.e., correlation), absolute performance (i.e., NMAE), and performance against the best policy (i.e., regret). All three metrics captured three different and important aspects of the learning problem, providing a lot of insight.

**Weaknesses:**

- This paper needs more analysis regarding why certain features work better for certain settings. At the moment, if I implement this idea, I would have to enumerate all possible pairs.
- The authors only compare to 1 baseline algorithm that only depends on the actions of the policies in the offline dataset. Meanwhile, the proposed method uses the state-action in the offline dataset to learn the successor feature components. I don't know if the performance increase of the baseline is due to the proposed idea or the additional information the proposed idea has access to.

**Questions:**

- If you had online policy trajectories from behavior policies, how would that affect the proposed idea?
- How did you find the best policy to compare against for the regret metric? Was the best policy feature-dependent or feature-agnostic?
- Why is the correlation metric related to how many evaluations on the reboot are required to find the best policy? I thought correlation was the relationship between the set of predicted values and ground-truth values. This relationship could be arbitrarily bad; no matter how many evaluations you do, the underlying feature may not provide a reasonable signal that relates to the ground truth.
- Can I assume that correlation means relative performance (i.e.,  the ordering of values between the prediction and ground are the same), while NMAE is absolute performance (i.e., the predicted and ground values are exactly the same)?
- What is significant of NMAE? The performance values of Table-1 and Table-2 imply that correlation indicates regret. This means that representations with low regret have a high correlation, but NMAE does not have a relationship to regret.
- Why is action representation the only baseline applicable baseline? The underlying data has states and actions. Would it be unfair to condition a baseline on both state-action pairs?
- The discussion from (i), (ii), and (iii) in the results section is confusing. In results (ii), the authors raise the point that NMAE is better, but in (iii), the authors raise the point that their approach is better in regret. What metric is the most important across these metrics presented?


Missing cites:

- Original Successor feature paper:  Improving generalization for temporal difference learning: The successor representation by Dayan, et al. 1993.
- Successor Feature Representations by Reinke et al 2023
- Successor Feature Sets: Generalizing Successor Representations Across Policies by Brantley et al. 2021
- Successor Features Combine Elements of Model-Free and Model-based Reinforcement Learning by Lehnert et al 2020

---

> ### Author Response · Authors · 2023-11-16
>
> Dear reviewer *81Fm*, we appreciate the insightful feedback and reference to missing citations, which helped improve our manuscript. We address the question about online policy representation and metrics in the general comment, and next, we address the remaining concerns below.
>
> **Better analysis of the features**: we present our intuition regarding the features to adopt in paragraph (iv) in the Appendix. We compare CLIP and VIT across all environments. Tables 1 and 2 show a correlation between the choice of the feature encoder and policies’ true return values. CLIP performs better than VIT in Insert Gear (Sim), Insert Gear (Real), and Metaworld. Table 4 reports policy performances in these environments, showing they are the hardest to solve, as the standard deviation among policies is high. These results demonstrate that CLIP is robust when the task is hard, and most of the policies fail to solve it. We stress that $\pi$2vec might be especially useful for this use case, as corresponds to a scenario where there is an ongoing research effort to train policies for a particular task on a real robotic platform. On the other hand, VIT is the best option when the standard deviation of policy returns are low or negligible (e.g., Kitchen and RGB stacking), making this feature most suited for ranking policies that consistently solve the task.
>
> **Baseline comparison**: We point out that the action representations are extracted by running the historical policies on the set of canonical states to compute their actions. We adopt the same set of canonical states for $\pi$2vec and action representations. In this regard, the baseline and $\pi$2vec have access to the same data, and the baseline uses state information implicitly by deciding which actions should be compared. Active Offline Policy Selection (AOPS) [1] stands alone as a notable work that delves into policy representation from offline data with the task of deciding which policies should be evaluated in priority to gain the most information about the system. We argue that this is a strong baseline, as demonstrated in [1] "actions baseline" proves to be insightful for understanding policy behavior (Figure 3 in [1]) and predicting policy performance (Figure 5 in [1]).  We are not aware of any previous work that addresses this particular problem by bringing foundation model representations into offline policy representation and, in particular, any other prior method that would use states in a more explicit way.
>
> **Finding the best policy**: when reporting average results across tasks–Metaworld and Kitchen–we cross-validate $\pi$2vec with various features on the historical policies for each task. Next, for each task, we pick $\pi$2vec with the feature representation that achieves the lowest regret. We report the average of the per-task best representations as Best-* in Table 2.
>
> [1] Konyushova, Ksenia, et al. "Active offline policy selection." Advances in Neural Information Processing Systems 34 (2021): 24631-24644.

---

### Author Response · Authors · 2023-11-16
**Paper revision**

For the revised version, our main updates are summarized below and marked in blue in the manuscript:
* We introduce additional experiments showing robustness to canonical state coverage (paragraph ix in additional experiments, Appendix) and method performance with online data (paragraph x in additional experiments, Appendix).
  + In answering the reviewer's *4YPE* question, we show that $\pi$2vec is robust even when using only the first state of demonstrations as canonical states for $\pi$2vec representations. By relying on the successor feature framework for estimating the discounted sum of features over the entire trajectory, we show that $\pi$2vec represents policies’ behavior as vectors without relying on the specific content of the canonical states.
  + We follow reviewers' *eE8e* and *81Fm* suggestions and estimate $\pi$2vec performance when adopting online data for policy representations. $\pi$2vec’s representations improve due to better coverage of the training dataset.
* To clarify the importance of the proposed setting, we explain the use cases for $\pi$2vec and add a paragraph on how RL policy developers could use the approach in the introduction of the revised manuscript.
* We introduce a Metrics Section in the Appendix to discuss metrics choice and interpretation further, as suggested by reviewers *81Fm* and *Kqe9*suggested.
We add additional citations mentioned by *81Fm*.

---

### Author Response · Authors · 2023-11-16
**General Comment to all reviewers (i)**

We thank all reviewers for their thorough reviews of our paper and their acknowledgment of our approach in terms of novelty (*81Fm*, *4YPE*, *eE8e*, *Kqe9*), strong empirical evaluation (*81Fm*, *4YPE*, *eE8e*), and the importance of the task (*81Fm*,*4YPE*).
We hope our answers clarify the main concerns. Let us know if you have additional questions.

## Use case (*Kqe9*, *4YPE*):

Our proposed method $\pi$2vec offers advancements during the active policy development stage, particularly when evaluating trained policies is resource-intensive.
This is particularly relevant in reinforcement learning projects, where researchers often face the challenge of efficiently evaluating and validating improvements in policy architecture or training procedures. This process can be particularly costly and time-consuming when involving real robotic systems (e.g., *Insert Gear real* task in this paper).

What sets $\pi$2vec apart is its ability to predict the performance of untested policies based on the data from already evaluated ones. $\pi$2vec enables researchers to make more informed decisions regarding which new policies to prioritize for real-world testing or to identify and discard less promising options early in the development process.
This setting with gradual policy development implies that the new policies are related to the existing ones and the data coverage is comprehensive enough to allow $\pi$2vec to make accurate predictions about the performance. Thus, we are able to optimize the use of resources and time in the RL policy development cycle.

## $\pi$2vec from online data (*eE8e*, *81Fm*):
In this paper, we are interested in a setting where running policies in a real environment is time-consuming and can lead to faults and damages and, thus, researchers are interested in training and evaluating policies based on the offline data as much as possible. If we had a chance to collect online data, then it could be used for policy evaluation and further policy training. Nonetheless, to answer a question from reviewers *eE8e* and *81Fm*, in the next experiment, we test $\pi$2vec for representing policies from online data. For each policy $\pi$, we collect a dataset of trajectories by deploying the policy in the environment. Next, we train $\psi_\pi$ on the dataset of $\pi$'s trajectories and compute its $\pi$2vec's representation. The following Table reports results when training $\pi$2vec on online data on Insert Gear (sim) task. We show that $\pi$2vec’s performances improve with respect to the offline counterpart (reported in Table 1 in the main paper).This result is expected: a better dataset coverage leads to improved results, as we also discuss in paragraph *(v)* in the Appendix. In particular, $\pi$2vec’s regret@1 improves by 0.15 when using TAP features and 0.9 when using CLIP features. The results for the offline data hold for the online data as well: geometrical TAP features are better at correlation and NMAE, which is especially useful for Offline Policy Evaluation. On the other hand, $\pi$2vec with CLIP features archives the lowest regret@1, which is ideal for Offline Policy Selection. We updated the Appendix with this additional result.
| Feature |	NMAE | Correlation | Regret@1 |
|---------|---------|-------------|----------|
| Actions |   0.174 |   	0.650 |	0.427 |
| CLIP	| **0.158** |   	0.627 |	0.288 |
| Random  |   0.348 |   	0.425 |	0.302 |
| TAP 	|   0.172 | 	**0.686** |  **0.205** |

---

> ### Author Response · Authors · 2023-11-16
> **General comment to all reviewers (ii)**
>
> ## Metrics ( *81Fm* and *Kqe9*):
> We adopt three standard metrics that are commonly used for Offline Policy Evaluation [1]. We report the equations below and update the Appendix of the manuscript accordingly.
> Normalized Mean Absolute Error: This metric is defined as the difference between the value and estimated value of a policy:
> \begin{equation}
> \textrm{NMAE} = | V^\pi - \hat{V}^\pi |
> \end{equation}
>
> Where $V^\pi$ is the true value of the policy, and $\hat{V}^\pi$ is the estimated value of the policy.
>
> Regret@1 is the difference between the value of the best policy in the entire set and the value of the best predicted policy. It is defined as:
>
> \begin{equation}
> \textrm{Regret @ k} = \max_{i \in 1:N} V^\pi_{i}  -  \max_{j \in (1:N)}  V^\pi_{j}.
> \end{equation}
>
>
> Rank Correlation (also Spearman's $\rho$) measures the correlation between the estimated rankings of the policies’ value estimates and their true values. It can be written as:
>
> \begin{equation}
> \textrm{RankCorr} = \frac{\textrm{Cov}(V^\pi_{1:N}, \hat{V}^\pi_{1:N})}{ \sigma(V^\pi_{1:N}) \sigma(\hat{V}^\pi_{1:N})}
> \end{equation}
>
> ## When is each of the metrics applicable? (*81Fm*)
> Correlation and regret@1 are the most relevant metrics for evaluating $\pi$2vec on Offline Policy Selection (OPS), where the focus is on ranking policies based on values and selecting the best policy [1].
> Regret@1 is commonly adopted in assessing performances for Offline Policy Selection, as it directly measures how far off the best-estimated policy is to the actual best policy.
> Correlation is relevant for measuring how the method ranks policies by their expected return value. By relying on methods that achieve higher correlation and thus are consistent in estimating policy values, researchers and practitioners can prioritize more promising policies for online evaluation.
> On the other hand, NMAE refers to the accuracy of the estimated return value. NMAE is especially significant when aiming at estimating the true value of a policy and is suited for Offline Policy Evaluation (OPE), where we are interested in knowing the values of each policy. We assess $\pi$2vec’s representation in both settings, showing that $\pi$2vec consistently outperforms the baseline in both metrics. We improve the discussion on metrics in the Appendix of the manuscript.
>
>
> [1] Justin Fu, Mohammad Norouzi, Ofir Nachum, George Tucker, Ziyu Wang, Alexander Novikov, Mengjiao Yang, Michael R Zhang, Yutian Chen, Aviral Kumar, et al. Benchmarks for deep off-policy evaluation. arXiv preprint arXiv:2103.16596, 2021

---

### Meta-Review · Area_Chair_RoBW · 2023-12-19

**Metareview:**

The paper presents an approach for representing reinforcement learning policies using successor features and foundation models for off-policy evaluation. Reviewers appreciated the method's novelty, thorough empirical investigation across a variety of domains (including real-world robots), and clarity. There were concerns about the lack of theoretical analysis, limited baseline comparisons, and the method's restriction to offline settings. Despite these issues, I recommend acceptance as the method could be of value to the community.

**Justification For Why Not Higher Score:**

Limited baseline comparisons, lack of analysis and investigation for why the method works.

**Justification For Why Not Lower Score:**

Interesting and novel combination of tools and empirical evaluation across a range of non-trivial domains.

---

### Decision · Program_Chairs · 2024-01-16

Accept (poster)